# Combined Blockade of TIGIT and PD-L1 Enhances Anti-Neuroblastoma Efficacy of GD2-Directed Immunotherapy with Dinutuximab Beta

**DOI:** 10.3390/cancers15133317

**Published:** 2023-06-23

**Authors:** Nikolai Siebert, Maxi Zumpe, Christian Heinrich Schwencke, Simon Biskupski, Sascha Troschke-Meurer, Justus Leopold, Alexander Zikoridse, Holger N. Lode

**Affiliations:** Department of Pediatric Oncology and Hematology, University Medicine Greifswald, 17475 Greifswald, Germany

**Keywords:** neuroblastoma, immunotherapy, NK cells, dinutuximab beta, TIGIT, PD-L1, MDSCs

## Abstract

**Simple Summary:**

To further improve the antitumor efficacy of GD2-directed immunotherapies against high-risk neuroblastoma (NB), we investigated, in the present work, a combinatorial immunotherapy, with the chimeric antibody (Ab) dinutuximab beta (DB) and the double immune checkpoint blockade of T-cell immunoreceptor with immunoglobulin and ITIM domain (TIGIT) and programmed cell death ligand-1 (PD-L1). ADCC by DB resulted in a strong tumor cell lysis but induced the expression of PD-L1 and TIGIT on effector cells including NK cells. Immunotherapy with DB, combined with either anti-TIGIT or anti-PD-L1 Ab, effectively inhibited tumor growth and improved the survival of the tumor-bearing mice. The superior effects were observed when DB was combined with the double immune checkpoint blockade, thus presenting a new effective combinatorial treatment strategy against high-risk NB.

**Abstract:**

Immunotherapies against high-risk neuroblastoma (NB), using the anti-GD2 antibody (Ab) dinutuximab beta (DB), significantly improved patient survival. Ab-dependent cellular cytotoxicity (ADCC) is one of the main mechanisms of action and it is primarily mediated by NK cells. To further improve antitumor efficacy, we investigated here a combinatorial immunotherapy with DB and the double immune checkpoint blockade of T-cell immunoreceptor with immunoglobulin and ITIM domain (TIGIT) and programmed cell death ligand-1 (PD-L1). The effects of ADCC, mediated by DB against NB cells on NK-cell activity, and the expression of TIGIT and CD226 and their ligands CD112 and CD155, as well as of PD-1 and PD-L1 on NB and effector cells, were investigated using flow cytometry. ADCC was assessed with a calcein-AM-based cytotoxicity assay. The efficacy of a combinatorial immunotherapy with DB, given as a long-term treatment, and the double immune checkpoint blockade of TIGIT and PD-L1 was shown using a resistant murine model of NB, followed by an analysis of the tumor tissue. We detected both TIGIT ligands, CD112 and CD155, on all NB cell lines analyzed. Although ADCC by DB resulted in a strong activation of NK cells leading to an effective tumor cell lysis, a remarkable induction of PD-L1 expression on NB cells, and of TIGIT and PD-1 on effector cells, especially on NK cells, was observed. Additional anti-TIGIT or anti-PD-L1 treatments effectively inhibited tumor growth and improved survival of the mice treated with DB. The superior antitumor effects were observed in the “DB + double immune checkpoint blockade” group, showing an almost complete eradication of the tumors and the highest OS, even under resistant conditions. An analysis of tumor tissue revealed both TIGIT and TIGIT ligand expression on myeloid-derived suppressor cells (MDSCs), suggesting additional mechanisms of protumoral effects in NB. Our data show that the targeting of TIGIT and PD-L1 significantly improves the antitumor efficacy of anti-GD2 immunotherapy, with DB presenting a new effective combinatorial treatment strategy against high-risk tumors.

## 1. Introduction

Neuroblastoma (NB) is an aggressive pediatric cancer with a survival rate of only 50% in the high-risk group [1]. Recently, a GD2-directed immunotherapy, with the chimeric anti-GD2 antibody (Ab) dinutuximab beta (DB), has improved treatment efficacy by 15% [2]. The clinical activity of DB and other anti-GD2 Ab relies on the high expression of the GD2 antigen on NB cells and on the activity of natural killer (NK) cells, which are the major effector cell population mediating ADCC. Despite the promising results of the GD2-directed immunotherapies, one third of patients still relapse, thus emphasizing the need for improvement in the treatment’s efficacy. In this context, either the application of immune stimulating agents to further increase NK-cell activity against tumor cells, or the blockade of inhibitory pathways that negatively regulate immune responses, or both, are promising strategies. We have previously shown that the usage of the immune-stimulating immunocytokine FAP-IL-2v specifically binding fibroblast activation protein α (FAP) on tumor cell stroma and tumor-infiltrating MDSCs, via the FAP moiety of the immunocytokine, and that selectively activating NK cells via the mutated variant of IL-2, linked to the FAP fragment without the induction of Treg, resulted in a significant improvement in immunotherapy with DB [3]. These effects are in marked contrast to that of the “non-mutated” cytokine IL-2 broadly used in GD2-directed treatment regimens against high-risk NB. Recently, no treatment benefit of the addition of cytokine has been shown in the high-risk NB patients receiving DB in combination with the wild type (non-modified) IL-2 [4]. While NK cells were effectively induced, showing around 3-fold higher levels compared to the baseline (prior to start of treatment) in the patients additionally receiving IL-2, the concomitant increase in regulatory T cells (Treg), showing almost 20-fold higher levels, was also detected in these patients [5], thus suggesting the missing treatment advantage of IL-2.

The second treatment option to improve antitumor efficacy is a reduction in the inhibiting pathways, especially those that negatively regulate NK-cell activity. We previously showed a significant increase in anti-NB effects when the DB-based immunotherapy was combined with anti-PD-1 Ab [6]. These promising results in our preclinical study were clearly confirmed in high-risk patients [7], emphasizing the efficacy of immune checkpoint blockades in high-risk NB. Although NB belongs to the “cold” tumors (low levels of neo-antigens, low MHC-I expression) and immune checkpoint inhibitors would not be effective as single-agent treatments, we previously showed a significant induction of both PD-L1 and PD-1 expression on NB and effector cells, respectively, after the induction of anti-GD2 ADCC with DB [6]. These data clearly show a rationale to combine therapeutic Ab with immune checkpoint inhibitors against NB. In the context of the reduction in immune-inhibiting factors in NB, we further confirmed the negative role of MDSCs, whereby their reduction was effective against NB, even as a monotherapy; however, the strongest antitumor effects were shown when combined with DB [8].

In the present work, we investigated the role of a further immune checkpoint that is known to modulate NK-cell activity, namely TIGIT [9]. By interacting with its ligands, CD155 and CD112, TIGIT inhibits NK-cell- and CD8-T-cell mediated cytotoxicity, thus maintaining self-tolerance during immune responses. TIGIT expression is induced as a result of effector cell activation and has higher binding affinity to CD112 and CD155 compared to the immune stimulating receptor, CD226. As expression of the TIGIT ligands, CD112 and CD155, has been shown in different cancers [10], including NB [11], the blockade of TIGIT presents a further attractive therapeutic strategy, especially in the context of Ab-based immunotherapies, and has been already reported in experimental models to promote an NK-cell-dependent antitumor response in vitro and in vivo [10]. Importantly, in one recent study of melanoma, it was shown that the TIGIT blockade primarily acts on NK cells to inhibit tumor growth, as NK-cell depletion resulted in an abrogation of the antitumor effects of the TIGIT blockade [12]. Therefore, the blockade of TIGIT presents a further attractive therapeutic strategy, especially in the context of Ab-based immunotherapies.

Since treatment with DB leads to the induction of the PD-1/PD-L1 pathway, and PD-L1 expression has been shown to be coupled with TIGIT [13], we hypothesized that a dual blockade of PD-1/PD-L1 and TIGIT, in combination with DB, might be more effective than a single blockade. Indeed, the efficacy of the anti-TIGIT + anti-PD-1/PD-L1 combination has already been shown in different cancers, excepting NB [10].

In the present study, we addressed whether the TIGIT blockade, in combination with anti-PD-L1 treatment, improves the antitumor efficacy of the chimeric anti-GD2 Ab DB against NB. We first assessed the expression of CD226 and TIGIT, as well as their ligands, CD112 and CD155, by effector cells with focus on NK cells, as well as on primary and well-known NB cells, respectively. We then investigated the effects of ADCC mediated by DB against GD2-positive NB cells on the expression of CD226, TIGIT and PD-1 by NK cells, and on the expression of CD112, CD155 and PD-L1 by tumor cells. Furthermore, we investigated the antitumor efficacy of the combined immunotherapy with DB, TIGIT and PD-L1 blockades in vitro and in vivo using our resistant murine tumor model of NB, followed by an evaluation of tumor infiltrating effector cells with a special focus on MDSCs.

## 2. Materials and Methods

### 2.1. Ethic Statement

The experiments using samples collected from human participants were in accordance with the ethical standards of our institutional and national research committee, as well as with the 1964 Helsinki Declaration and its later amendments or comparable ethical standards. Informed consent was obtained from every individual participant. All investigations including experimental animals were approved by the animal welfare committee (Landesamt für Landwirtschaft, Lebensmittelsicherheit und Fischerei Mecklenburg-Vorpommern, LALLF M-V/7221.3-1-007/22) and were supervised by the local commissioner for animal welfare at the University Medicine Greifswald, representing the Institutional Animal Care and Use Committee (IACUC).

### 2.2. Cell Cultivation

For evaluation of expression of the TIGIT ligands, CD112 and CD155, on tumor cells, following NB cell lines were analyzed: LAN-1, CHLA-20, CHLA-90, CHLA-136 and COG-N-519. Additionally, we analyzed in-house-established HGW-1, HGW-3, and HGW-5 cell lines that were established using tumor samples originating from high-risk NB patients [14]. The cell lines, excepting LAN-1, were cultivated with IMDM (PAN-Biotech GmbH, Aidenbach, Germany) supplemented with stable glutamine (4 mM; Fisher Scientific, Hampton, NH, USA), 1× ITS (insulin/transferrin/selenium; Capricorn Scientific GmbH, Ebsdorfergrund, Germany), 1× non-essential amino acids (Capricorn Scientific GmbH, Ebsdorfergrund, Germany), penicillin (50 U/mL) and streptomycin (0.05 mg/mL) (0.5× P/S; PAN-Biotech GmbH, Aidenbach, Germany) and 20% FBS Good (PAN-Biotech GmbH, Aidenbach, Germany). For LAN-1, we used RPMI 1640 (Capricorn Scientific GmbH, Ebsdorfergrund, Germany); 2 mM stable glutamine, 0.5× P/S and 10% Sera Plus (PAN-Biotech GmbH, Aidenbach, Germany).

For tumor induction in vivo, murine NB cells NXS2-HGW were used [6,15]. The cells were cultivated in DMEM (2 mM stable glutamine, 0.3× P/S and 10% FBS Good).

To show effects of ADCC mediated by DB on expression of TIGIT, CD226 and PD-L1 on effector cells, as well as effects of immune checkpoint blockade on ADCC against NB cells mediated by DB and on activation of NK cells, leukocytes of healthy donors (effector cells) were used. Prior to analysis, the cells were cultivated using RPMI 1640 supplemented with 2 mM stable glutamine, 0.5× P/S and 10% Sera Plus.

### 2.3. Flow Cytometry

To examine baseline expression of the receptors CD226, TIGIT and PD-1 on effector cells as well as their respective ligands, CD112, CD155 and PD-L1, on tumor cells, flow cytometry analysis was used. Moreover, we investigated expression of both receptors on NK cells as a major cell population mediating ADCC. To investigate whether ADCC, mediated by DB, against GD2-positive tumor cell changes’ expression of CD226, TIGIT and PD-1 on effector cells, with focus on NK cells, and on CD112, CD155 and PD-L1 on tumor cells, we performed flow cytometry analysis after induction of ADCC using the GD2-positive NB cells LAN-1, leukocytes of a healthy donor and DB, as described in “ADCC” section.

Additionally, we evaluated effects of DB-ADCC on CD107a expression levels by NK cells as a functional marker for the identification of their activity.

For detection of CD112, CD155, CD226 and TIGIT, anti-CD112 (APC-conjugated; Miltenyi Biotech, Teterow, Germany), anti-CD155 (PE-Vio 770-conjugated; Miltenyi Biotech, Teterow, Germany), anti-CD226 (VioBlue-conjugated; Miltenyi Biotech, Teterow, Germany) and anti-TIGIT Ab (PerCP-Cy5.5-conjugated; BioLegend, San Diego, CA, USA; Brilliant Violet 421-conjugated; BioLegend, USA) were used. The expression of PD-1 and PD-L1 was measured using anti-PD-1 (Brilliant Violet 421-conjugated; BioLegend, San Diego, CA, USA) and anti-PD-L1 Ab (PE-conjugated; BioLegend, San Diego, CA, USA), respectively. To detect NB cells, the Alexa Fluor647-labeled anti-GD2 chimeric Ab DB was utilized. NK cells were defined as GD2−/CD45+/CD56+ cells (PE-Cy7-conjugated anti-CD45 Ab; Miltenyi Biotech, Teterow, Germany, APC-Vio.770-conjugated anti-CD56 Ab; Miltenyi Biotech, Teterow, Germany). To detect CD107a, anti-CD107a Ab (Vio Bright-conjugated; Miltenyi Biotech, Teterow, Germany) was used.

To evaluate TIGIT and CD155 expression in murine tumor tissue, Tumor Dissociation Kit (Miltenyi Biotech, Teterow, Germany) was used to prepare single-cell suspensions according to the manufacturer’s instructions. Thereafter, flow cytometry analysis was performed. For this, anti-CD3 Ab (APC-Vio770-conjugated), anti-CD45 Ab (VioGreen-conjugated), anti-CD335 Ab (PE-conjugated), anti-CD155 (PE-Vio770-conjugated), anti-TIGIT (Brilliant Violet 421-conjugated; all from BioLegend, San Diego, CA, USA) and anti-GD2 Ab (Alexa Fluor 647-conjugated) were used. 

For detection of MDCSs in mouse tumors, following Ab were used: anti-CD45, anti-TIGIT and anti-CD155 Abs (see above), anti-CD11b (Vio Brght FITC-conjugated), anti-Ly6G (APC-conjugated, all from Miltenyi Biotech, Teterow, Germany) and anti-Ly6C Ab (APC-Cy7-conjugated; BioLegend, San Diego, CA, USA). 

FcR Blocking Reagent (Miltenyi Biotech, Teterow, Germany) was used, to exclude unspecific binding of the detection Ab to Fc receptor-expressing cells. Data analysis was performed using FlowJo V10 software (Ashland, OR, USA). To discriminate between dead and viable cells in the samples, 4′,6-diamino-2-phenylindole solution (DAPI, 13.3 µg/mL), 7-Aminoactinomycin D (0.825 µg/mL) or propidium iodide (0.825 µg/mL) was added prior to acquisition with the BD CANTO II cytometer and FACS Diva v8.0.2 software (BD Biosciences, Heidelberg, Germany).

To determine the potential background caused by nonspecific Ab binding, a respective isotype control (ITC) was utilized for every primary Ab. After exclusion of non-specific binding by means of isotype controls, we used unstained cells or “fluorescence minus one” (MFO) controls for mono- or multi-staining, respectively, in the following gating strategy.

### 2.4. ADCC

To investigate effects of TIGIT- and PD-1/PD-L1 blockade on the cellular cytotoxicity of effector cells against GD2-expressing NB cells, mediated by DB (ADCC), we used the previously reported calcein-AM-based cytotoxicity assay [14]. Briefly, prior to induction of ADCC, effector cells of healthy donors (leukocytes) were cultivated for 24 h with RPMI 1640 supplemented with 2 mM stable glutamine, 0.5× P/S, 10% Sera Plus and IL-2 (100 IU/mL). To induce ADCC against NB cells, the GD2-positive NB cells LAN-1 (5000 cells/well), labeled with Calcein, the anti-GD2 Ab DB (10 µg/mL) and leukocytes, at an effector-to-target-cell ratio of 10:1, were incubated for 4 h, followed by ADCC measurement. To show that ADCC was GD2-specific, the anti-idiotype Ab ganglidiomab was used [16]. To assess DB-independent cytotoxicity (AICC, antibody-independent cellular cytotoxicity), leukocytes were incubated with tumor cells without DB. Untreated tumor or effector cells served as controls.

To show effects of ADCC on expression of CD226, TIGIT and PD-1 on effector cells, as well as CD112, CD155 and PD-L1 on tumor cells, GD2-positive NB cells LAN-1 and effector cells from healthy donors were incubated with 10 µg/mL DB for 24 h, at an effector-to-tumor-cells ratio of 10:1, prior to flow cytometry analysis. Additionally, ADCC impact on CD107a expression levels by NK cells was assessed as a functional marker for the identification of NK-cells activity.

### 2.5. Evaluation of Antitumor Efficacies of DB-Based Immunotherapies In Vivo

Prior to investigation of antitumor efficacy of the combined immunotherapy with DB, and the double immune checkpoint blockade of TIGIT and PD-L1, we established a more clinically relevant long-term treatment schedule for DB immunotherapy. For that, we used a more resistant version of our syngeneic tumor model allowing evaluation of combinatorial immunotherapies, as described previously [3]. In contrast to the original protocol (DB was given from day 4 after tumor cell injection, tumors were not yet measurable), in the more resistant model version, DB treatments were started in a later tumor growth phase, after development of measurable tumors showing volumes of about 100 mm^3^. In this work, we further adapted this protocol. For that, we compared tumor growth in mice treated according to our previously established short-term treatment protocol (DB, 3 mg/kg bw/day, i.p., five consecutive days) [3] with a long-term treatment protocol (DB, 3 mg/kg bw/day, i.p., every second day for ten days) that has been shown in clinic to be associated with improved side effect profiles compared to the short-term treatments [17]. DB treatments were started in both experimental groups after tumors developed volumes of about 100 mm^3^. A/J mice (Charles River Laboratories, Sulzfeld, Germany, 9 weeks old) were granted a two-week acclimatization time. Mice groups (max. 6 animals) were accommodated in standard animal laboratories (20 ± 2 °C RT, 60 ± 20% humidity, ad libitum access to water and standard laboratory chow, 12 h light/dark cycle). NB cells (2 × 10^6^) were injected on the left ventral flank subcutaneously. The treatments were started when tumors developed a tumor size of 100 mm^3^. Assessment of tumor and/or treatment burden parameters [6] was carried out every two days after tumor cell injection. From day 8, these analyses were performed daily. Tumor volume was calculated as follows: length × width × height)/2. Mice were sacrificed when tumors exceeded 750 mm^3^. For those mice that were killed ahead of schedule, the tumor volume calculated at the last measurement was included in the assessment of the average volumes of the respective group at the subsequent time points.

After the successful establishment of long-term immunotherapy with DB, we used this new protocol in the following in vivo experiments for the evaluation of antitumor efficacy of the DB immunotherapy, in combination with blockade of TIGIT and PD-L1. For that, mice were randomized prior to tumor cell injection. DB was given every second day (days 1, 3, 5, 7 and 9) according to the long-term treatment protocol, as described above. The immune checkpoint inhibitors of the murine anti-TIGIT Ab (10 mg/kg bw/day; F. Hoffmann-La Roche Ltd., Basel, Switzerland/ Genentech, Inc., MS, USA) and the murine anti-PD-L1 Ab (10 mg/kg bw/day; F. Hoffmann-La Roche Ltd., Basel, Switzerland/ Genentech, Inc., MS, USA) were given i.p. on days 1, 4, 5, 8, 11, 15, 18 and 20.

### 2.6. Assessment of DB Serum Levels in Treated Mice

For detection of DB in mice of the “long-term treatment” group, blood samples were collected on days 1, 2, 3 and 4 after end of DB treatment. The serum concentration of DB was determined using the triple-ELISA strategy, as previously described [18,19]. We used, as a capture Ab, the anti-idiotype Ab ganglidiomab [16]. Briefly, mice serum samples were first analyzed by the “low sensitivity” ELISA, allowing detection of DB between 3.0 and 25 µg/mL. Thereafter, samples showing DB levels lower than 3 µg/mL were re-analyzed with the “intermediate sensitivity” ELISA, with detection range of 0.5–3.1 µg/mL. Finally, samples showing DB concentrations below 0.5 µg/mL were re-analyzed again using the ELISA with the highest sensitivity (detection range of 0.058–1.0 µg/mL).

### 2.7. Statistics

For the statistical analysis, SigmaPlot software (Version 13.0, Jandel Scientific Software) was used. Firstly, the acquired data sets were tested for normal distribution. Then, dependent on the outcome of the analysis, either the Mann–Whitney U-test, or Students *t*-test in the case of the assumption of normality, and analysis of variance (ANOVA) for comparison of more than two groups regarding the significance of a metric trait, were used. The data are presented as mean ± SEM (standard error of the mean). Overall survival (OS) was assessed with LogRank test, multiple comparisons were performed using the Holm–Sidak method for the post hoc testing. A *p* value of <0.05 was considered significant, <0.01 very significant and <0.001 highly significant.

## 3. Results

### 3.1. Evaluation of Expression of the TIGIT Ligands, CD112 and CD155, on NB Cells

Firstly, we analyzed the human NB cell lines LAN-1, CHLA-20, CHLA-90, CHLA-136 and COG-N-519 for the basal expression of the two TIGIT ligands, CD112 and CD155, by flow cytometry. Additionally, we assessed CD112 and CD155 on the in-house-established cell lines that originated from the tumor samples of high-risk NB patients (HGW-1, HGW-3 and HGW-5) [14].

All cell lines analyzed showed a clear signal for both ligands with different levels of expression (Figure 1), thus suggesting the role of the TIGIT pathway in NB.

### 3.2. Effects of ADCC Mediated by DB on Expression of TIGIT and CD226 on Effector Cells as Well as CD112 and CD155 on Tumor Cells

To investigate whether ADCC, mediated by DB against NB cells, impacts on the expression of TIGIT and CD226 on effector cells, with a special focus on NK cells as a major cell population mediating ADCC, as well as CD112 and CD155 on GD2-positive tumor cells (LAN-1), flow cytometry analysis of both effector and tumor cells, harvested after the induction of ADCC, was performed.

There was a trend of increasing expression of both ligands, CD112 and CD155, on tumor cells by DB-ADCC compared to the controls (Figure 2A). Similar results were observed in an analysis of the activating receptor CD226 on leukocytes and NK cells (Figure 2B,C). Importantly, an evaluation of TIGIT expression revealed significantly increased levels by the effector cells, and highly significant increased levels by NK cells of the ADCC group (Figure 2B,C) compared to the similar levels in the untreated and AICC samples, clearly showing ADCC-dependent effects mediated by DB on the expression of this inhibiting receptor.

Taken together, these data show that ADCC, mediated by DB against tumor cells, leads to a strong induction of the inhibitory receptor TIGIT on effector cells, especially on NK cells, thus further providing a rationale for blocking this inhibiting pathway for an improvement in the antitumor effects of DB-based immunotherapy.

### 3.3. Effects of ADCC Mediated by DB on NK-Cell Activity

Next, we investigated, by flow cytometry, whether ADCC mediated by DB activates NK cells, using CD107a as a functional marker for the identification of NK-cell activity. We additionally assessed, in our model, NK-cell numbers and numbers of the CD56dim NK cells relative to lymphocytes and all NK cells, respectively, that were found in the effector cell samples used in this work in order to validate our in vitro model.

Relative to all lymphocytes and NK cells found in the samples used, the frequencies of NK cells and the CD56dim NK-cell population were about 8% and 93%, respectively (Figure 3A), thus representing typical NK-cell numbers known for healthy donors. As expected, the induction of DB-ADCC against NB cells resulted in a strong activation of NK cells showing around a seven-fold increase in CD107a expression compared to the NK cells of the untreated control group (Figure 3B). Comparison of the ADCC and AICC experimental groups showed an about 2.5-fold difference with a significantly stronger tumor cell lysis mediated by DB. Additional incubation with the anti-idiotype Ab ganglidiomab completely abrogated ADCC, resulting in similar cytotoxicity levels compared to the Ab-independent control (AICC), thus clearly showing the GD2 specificity of the observed effect. Interestingly, compared to the untreated controls, the incubation of NB cells with leukocytes without DB (AICC) also resulted in a statistically significant increase in NK-cell activity, showing around 3-fold higher expression levels of CD107a, suggesting additional DB-independent effects on NK-cell activity by tumor cells only.

These data clearly show that ADCC, induced by the chimeric anti-GD2 Ab DB, increases NK-cell activity against tumor cells, thus further confirming the role of NK cells in the context of Ab-based treatments.

### 3.4. Effects of ADCC Mediated by DB on Expression of PD-L1 on Tumor Cells and PD-1 on NK Cells

Since TIGIT blockade, as a single-agent treatment, has been shown to be insufficient in driving an effective antitumor response in mice with established tumors [20], a combination of anti-TIGIT treatments with other immune checkpoint inhibitors, especially those that block the PD-1/PD-L1 pathway, represents a more effective antitumor strategy. Thus, we next investigated the impact of ADCC mediated by DB on the expression of PD-1 by effector cells, especially NK cells, and of PD-L1 by tumor cells.

In line with our previous observations showing the DB-ADCC-mediated induction of the immune checkpoint PD-1/PD-L1 [6], the incubation of tumor cells with effector cells and DB (ADCC) resulted in a strong increase in PD-L1 on tumor cells (Figure 4A) and PD-1 on NK cells (Figure 4B). The GD2 specificity of the observed effects was confirmed by the additional incubation with the anti-idiotype Ab ganglidiomab. Here, the ADCC-dependent induction of both the receptor PD-1 on NK cells, and its ligand PD-L1 on tumor cells, was completely abrogated showing similar cytotoxicity levels compared to the controls (untreated and AICC).

These data confirm our previous observations showing the DB-ADCC-dependent induction of the inhibitory pathway PD-1/PD-L1 in NB. Importantly, NK cells, which are the major effector cells mediating ADCC, showed a strong immunotherapy-dependent induction of the inhibitory receptor PD-1, further underlining the rationale for blocking this immune checkpoint in the context of GD2-directed immunotherapies against NB.

### 3.5. Effects of TIGIT Blockade, in Combination with Blockade of the PD-1/PD-L1 Pathway, on ADCC Mediated by DB

Based on our results of the impact of DB-ADCC the expression of TIGIT, and of PD-1 on effector cells, especially on NK cells, and their respective ligands on tumor cells, we next addressed the question of whether the blockade of these immune checkpoints leads to a stronger activation of NK cells and, in turn, augments DB-ADCC. We additionally compared the effects of two Abs blocking the PD-1/PD-L1 pathway, namely anti-PD-1 Ab nivolumab and anti-PD-L1 Ab atezolizumab.

As shown in Figure 5A, the additional blockade of either PD-1 or PD-L1 did not change the expression of CD107a by NK cells showing similar levels compared to the DB-ADCC group without the blockade Ab. Although the differences between the groups were not statistically significant, we observed a tendency for CD107a expression to increase by NK cells after the blockade of TIGIT compared to the DB-ADCC. Interestingly, the double immune checkpoint blockade showed a similar activity as the “DB + anti-TIGT” group, suggesting the minor role of the PD-1/PD-L1 pathway for NK-cell activity in this experimental design.

Next, prior to the evaluation of the effects of the double immune checkpoint blockade on DB-ADCC, we assessed the effects of the blockade Ab on DB-independent tumor cell lysis (Figure 5B). As expected, ADCC was induced only when tumor and effector cells were treated with DB. Neither atezolizumab nor nivolumab nor anti-TIGIT Ab provoked the cytotoxic activity of effector cells against NB cells without DB (Figure 5B). And, finally, we evaluated the effects of immune checkpoint blockades on DB-ADCC. Interestingly, the additional treatment of tumor cells with anti-PD-1 and PD-L1 Ab only slightly increased ADCC mediated by DB (Figure 5C). Although, in contrast, the combination “DB + anti-TIGIT Ab” resulted in a stronger ADCC showing an almost 10% higher tumor cell lysis compared to the monotherapy with DB, the differences between the groups were not found to be statistically significant. Then, we combined two immune checkpoint blockades with DB. Since similar effects of atezolizumab and nivolumab on NK-cell activity and ADCC with DB were observed, we used the anti-PD-L1 Ab atezolizumab in further experiments. In contrast to the results of the analysis of NK-cell activation showing similar levels in “DB + TIGIT” and “DB + TIGIT + atezolizumab” groups, a combination of DB with the double blockade resulted in a further improvement in ADCC observed with DB in combination with anti-TIGIT Ab (Figure 5C). However, the statistical analysis of these data did not reveal any significances between the groups. Based on our observations that the incubation of effector and tumor cells with anti-TIGIT Ab did not show any direct cytotoxic effects on tumor cells (Figure 5B), the observed improvement in ADCC by the double blockade was due to immune checkpoint inhibition.

Together, DB induced very effective tumor cells lysis by effector cells that could be tendentially improved by the double blockade of TIGIT and PD-L1. We also observed a tendency of increasing NK-cell activity after the blockade of TIGIT.

### 3.6. Establishment of the Dinutuximab Beta Long-Term Treatment Model

It has been shown that, compared to the patients treated by a short-term infusion (100 mg/m^2^, 5 days), the long-term treatment with DB (100 mg/m^2^, 10 days) resulted in an effective reduction in neuropathic pain [21], a common side effect of GD2-directed immunotherapies. We first adapted our treatment schedule in vivo prior to the analysis of DB immunotherapy combined with TIGIT and PD-L1 blockades. The new treatment protocol with DB is shown in Figure 6. Additionally, we assessed DB serum levels after the end of the long-term treatment.

In line with the results from the clinical studies, we observed a similar antitumor efficacy of DB given as a long-term treatment compared to the short-term treatment protocol (Figure 6B). Importantly, we observed immunologically active serum levels of DB showing even four days after the end of DB long-term treatment levels of about 4 µg/mL (4.19 ± 2.72 µg/mL; Figure 6C).

These data clearly indicate that the long-term treatment with DB leads to an effective Ab accumulation in the circulation of the treated mice. Based on these results, we used the optimized treatment protocol in our further experiments.

### 3.7. Effects of TIGIT Blockade in Combination with Blockade of PD-L1 on Antitumor Efficacy of Immunotherapy with DB In Vivo

After the successful establishment of the long-term DB treatment protocol, we used it for the evaluation of the antitumor efficacy of DB immunotherapy combined with the double blockade of TIGIT and PD-L1. We started to treat mice showing tumors of approximately 100 mm^3^ volume, as depicted in Figure 7. Mice receiving a monotherapy with DB, or DB in combination with either anti-TIGT or anti-PD-L1 Ab, served as controls.

As expected, in contrast to the strong antitumor efficacy of immunotherapy with DB that were reported in our less resistant NB model [6], the mice treated with DB alone showed, under more resistant conditions, a continuous growth of tumors (Figure 7B), confirming our previous results [3]. In contrast, both combinations of DB, with either anti-TIGIT or anti-PD-L1 Ab, strongly inhibited tumor growth (Figure 7B). Interestingly, the combination of DB with PD-L1 blockade improved the anti-NB efficacy of DB immunotherapy for a longer time period compared to the “DB + anti-TIGIT” combination. A strong inhibition of tumor growth was observed between days 11 and 16, and between days 12 and 15, for the “DB + anti-PD-L1” and “DB + anti-TIGIT” experimental groups, respectively. Moreover, in contrast to the “DB + anti-PD-L1” mice, the differences in tumor growth between the DB and DB + anti-TIGIT experimental groups were found, from day 16, to be no longer statistically significant. Furthermore, despite the double treatments with DB and anti-TIGIT Ab, we observed in these mice the continuous growth of tumors from day 12; however, this was not as strong as in the mice of the DB single-agent group. In contrast to the antitumor effects of the DB + anti-PD-L1 therapy, these showed a continuous inhibition of tumor growth already from day 7 (Figure 7B). Although the statistical analysis of the data did not reveal any significances between the double treatment groups (DB + anti-TIGIT and DB + anti-PD-L1), probably due to the low number of experimental mice per a group, these results suggest that the DB combination with PD-L1 blockade is more effective against resistant NB compared to the combination DB + anti-TIGIT Ab.

Finally, as hypothesized, combinatorial immunotherapy with DB, and the dual blockade of TIGIT and PD-L1, led to the strongest antitumor effects between the experimental groups, showing an almost complete eradication of the tumors in the treated mice (Figure 7B). The superior antitumor effects could be observed already in the first week after the start of treatment.

In line with this, our results of tumor growth could be completely confirmed by the evaluation of OS (Figure 7C). As expected, a single-agent treatment with DB showed, in such a resistant tumor model, a low OS probability (~10%). Additional immune checkpoint blockades improved the OS observed in the mice treated with DB alone, showing OS probabilities of about 60% and 80% in the mice of the “DB + anti-TIGIT” and “DB + anti-PD-L1” experimental groups, respectively (Figure 7C). Here, again, a combinatorial treatment with DB and PD-L1 blockade showed a superior antitumor efficacy compared to the DB and TIGT blockade combination. And, finally, the highest OS was found in the mice receiving three therapeutic agents, DB, anti-TIGIT and anti-PD-L1 Ab, clearly showing the benefit of the treatment with DB in combination with the double immune checkpoint blockade.

Together, the additional treatment of tumor-bearing mice with either anti-TIGIT or anti-PD-L1 augmented the antitumor effects of the anti-GD2 immunotherapy with DB, with superior effects in the “DB + anti-PD-L1” experimental group. Importantly, DB combined with a dual blockade of TIGIT and PD-L1 resulted in an almost complete eradication of the tumors and the highest OS probabilities, even under resistant conditions, thus representing a very promising treatment strategy against high-risk NB.

### 3.8. Analysis of TIGIT and CD155 Expression in Tumor Tissue

Finally, we analyzed the primary tumor samples collected from the experimental mice. We focused on TIGIT and CD155 expression on tumor and effector cells. As expected, we detected a clear expression of CD155 on tumor cells and of TIGIT on NK cells infiltrating tumor tissue (Figure 8A). Furthermore, we observed CD155 expression on MDSCs with higher levels on M-MDSCs compared to the second MDSC population, namely, PMN-MDSC (Figure 8B). These results underline an additional mechanism of inhibitory effects of these immune-suppressive cells in the tumor microenvironment. Surprisingly, we also detected the expression of TIGIT on both populations of MDCSs (Figure 8C), suggesting further mechanisms of their protumoral effects in NB.

These data confirm the expression of the TIGIT ligand CD155 on tumor cells and of TIGIT on NK cells infiltrating tumor tissue. We also found CD155 on both populations of MDCSs in tumor tissues and, importantly, we detected a clear TIGIT expression on these immune-suppressive cells, further emphasizing the rationale for blocking this pathway against NB.

## 4. Discussion

Although immunotherapeutic approaches with monoclonal anti-GD2 Ab significantly improves the survival of high-risk NB patients, about one-third of these patients still die [22], emphasizing the need for the establishment of more effective treatment strategies. It is known that ADCC is one of the major mechanisms of the antitumor effects of therapeutic Ab and that it is primarily mediated by NK cells; a more effective activation of this effector cell population could increase their antitumor cytotoxicity and thus improve patient outcomes. In the present study, we showed that ADCC, mediated by DB, effectively activated NK cells, resulting in a strong lysis of NB cells; however, it also strongly induced the immune checkpoint PD-1/PD-L1 on both tumor and effector cells [6] including NK cells (Figure 4). In line with these results, we observed the DB-ADCC-mediated induction of another immune checkpoint, that has been shown to regulate NK-cell activity [9], namely TIGIT. We found a highly significant increase in the expression of this inhibitory receptor on NK cells already 24 h after induction of DB-ADCC (Figure 2), confirming its role in NB. Moreover, we detected the expression of TIGIT ligands, CD112 and CD155, on all NB cells analyzed (Figure 1). These observations are in line with the results of other research groups reporting the expression of TIGIT ligands in different cancers, [10] including NB [11]. It has been shown that TIGIT is induced on effector cells as a result of their activation, a process maintaining self-tolerance during immune responses [13]. However, in cancer, increased TIGIT expression inhibiting NK-cell-mediated cytotoxicity can impair the efficacy of therapeutic agents. This suggests that the blockade of TIGIT presents a promising therapeutic strategy, especially in the context of Ab-based immunotherapies. Indeed, different anti-TIGIT Ab have been used in preclinical studies and, recently, in cancer patients [13]. Here, TIGIT blockades reduced tumor growth, promoting the infiltration of primary tumors by cytotoxic lymphocytes. Moreover, in patients with solid tumors, the interruption of TIGIT binding to its ligands showed controllable tolerance even when combined with anti-PD-1 Ab. An analysis of effector cells revealed increased NK-cell and cytotoxic lymphocyte activation, explaining the effects in these studies [13].

The mechanisms behind the tumor-dependent upregulation of immune checkpoints are not fully understood. Although some of the immune checkpoints are constitutively expressed on healthy cells, their expression can be strongly enhanced under stress conditions, especially during the transformation process [23]. We previously showed the direct role of CD11b-expressing effector cells in the DB-ADCC-mediated induction of the inhibitory pathway PD-1/PD-L1 [8]. In the last few years, many attempts were made to characterize and understand the role of CD11b-positive immune cells with immune suppressive characteristics. Most of these cells originate from a heterogeneous immature myeloid cell population, termed as myeloid-derived suppressor cells (MDSCs), that inhibit T- and, importantly, NK-cell activity and promote tumor growth, the development of metastases and contribute to resistance to immunotherapy [24]. We could show previously that a reduction in MDSCs significantly improved DB-immunotherapy and, interestingly, was effective even as a single-agent treatment [6,8]. Although single-agent blockades of immune checkpoints showed remarkable antitumor effects against different cancers, the induction of other immune checkpoints has been reported as a counter-reaction of such a treatment strategy. For instance, in the model of the head and neck squamous cell carcinoma, anti-PD-L1 treatment in vivo resulted in increased CD155 expression on MDSCs and, interestingly, vice versa, the blockade of TIGIT resulted in the upregulation of PD-L1 expression [25]. We could confirm, in the present study, the CD155 expression on MDCSs found in primary tumors. Surprisingly, besides ligand expression, we also observed on tumor-infiltrating MDCSs the expression of the receptor TIGIT (Figure 8), indicating an additional mechanism of action of this inhibitory pathway in NB. This might be one of the explanations as to why immune checkpoint blockades in the present work showed stronger effects in vivo compared to the in vitro assessments, lacking the immune modulating effects of TIGIT-positive myeloid cells. This is in line with the data of the study on the role of TIGIT on immune suppressive macrophages, another immune-suppressive cell population of the myeloid lineage [13]. Interestingly, TIGIT expression has been shown to be associated with the immunomodulation of the pro-inflammatory M1 macrophages into the immune-suppressive anti-inflammatory M2 macrophages secreting IL-10, thus supporting the development of the immune-suppressive environment [26]. Importantly, the TIGIT blockade resulted in the re-polarization of M2 toward the M1 phenotype, improving the phagocytosis of tumor cells [27,28].

Based on the fact that, besides TIGIT and its ligands, DB-ADCC also strongly induced the expression of the PD-1/PD-L1 pathway [6], and on the fact that the expression of both immune checkpoints has been shown to be coupled [13], we investigated here the anti-NB efficacy of a combinatorial treatment strategy with the chimeric anti-GD2 Ab DB in combination with the double immune checkpoint blockade. In contrast to our in vitro results, which showed only tendential improvement in DB-ADCC by the immune checkpoint blockades, in the tumor-bearing mice, we observed a marked enhancement of DB immunotherapy already by either anti-TIGIT or anti-PD-L1 treatments, with superior effects for the anti-PD-L1 treatment. These results show the efficacy of even one immune checkpoint blockade in combination with DB. The strongest effects on tumor growth and OS were observed in the mice receiving DB in combination with the double immune checkpoint blockade. Here, the mice showed an almost complete eradication of the tumors and the highest OS probabilities, even under resistant tumor conditions.

## 5. Conclusions

In summary, we report here the expression of the inhibitory receptor TIGIT and its ligands by NK and NB cells, respectively. We further showed the DB-ADCC-dependent induction of the expression of both immune checkpoints by tumor and effector cells. We also found the expression of TIGIT and its ligands on MDCSs in tumor tissue, suggesting the existence of further inhibitory mechanisms in NB. Finally, we showed, in a resistant syngeneic murine NB model, a significant improvement in the antitumor efficacy of the chimeric anti-GD2 Ab DB against NB by the additional blockade of the immune checkpoints TIGIT and PD-L1, providing an effective combinatorial treatment strategy against high-risk NB.

## Figures and Tables

**Figure 1 cancers-15-03317-f001:**
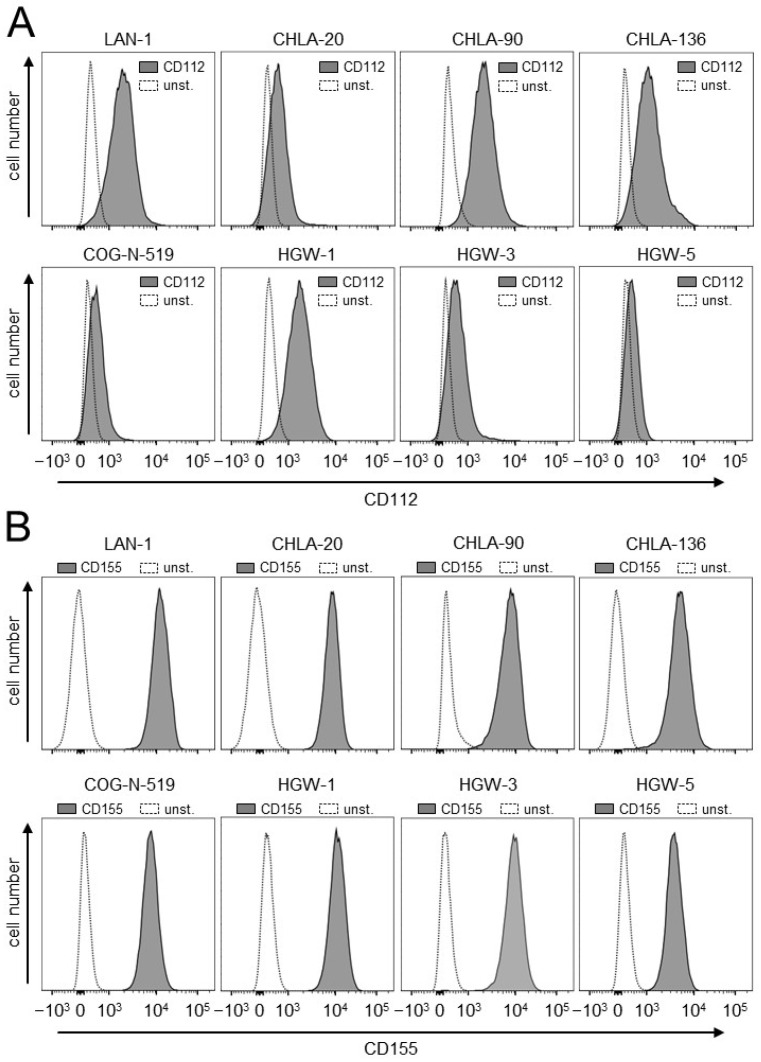
Analysis of the basal expression of the TIGIT ligands, CD112 and CD155, on NB cells. Analysis of CD112 (**A**) and CD155 (**B**) expression by human NB cells. Representative histograms of the flow cytometry analysis of CD112 and CD155 abundance on human NB cells (LAN-1, CHLA-20, CHLA-90, CHLA-136, COG-N-519, HGW-1, HGW-3 and HGW-5). Cells were stained with either anti-human anti-CD112 or anti-human anti-CD155 (black-filled curve). Unstained cells served as controls (unst., grey curve).

**Figure 2 cancers-15-03317-f002:**
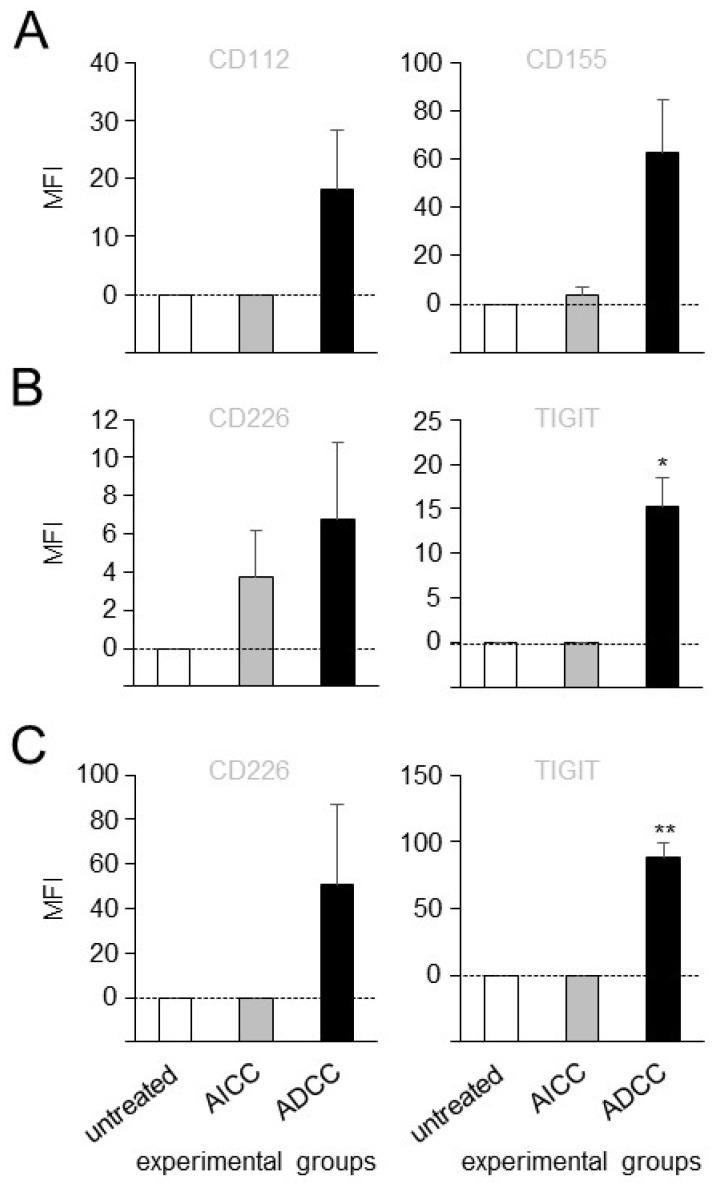
Impact of ADCC, mediated by DB, on expression of TIGIT and CD226 on effector cells, as well as CD112 and CD155 on tumor cells. (**A**) Effects of ADCC, mediated by DB, on expression levels of CD112 and CD155 on NB cells were assessed using flow cytometry. ADCC was induced by incubation of the GD2 expressing NB cells LAN-1 with leukocytes of different healthy donors and DB. Untreated NB cells (untreated) and NB cells incubated with effector cells without DB (AICC) served as controls. Effects of ADCC mediated by DB on expression levels of CD226 and TIGIT on leukocytes (**B**) and NK cells (**C**). Data are presented as a difference of gMFI (MFI), that was calculated according to the formula: experimental gMFI—gMFI of untreated controls. Data are shown as mean values ± SEM of at least 3 independent experiments. *t*-test; * *p* < 0.05 vs. untreated and AICC, ** *p* < 0.01 vs. untreated and AICC.

**Figure 3 cancers-15-03317-f003:**
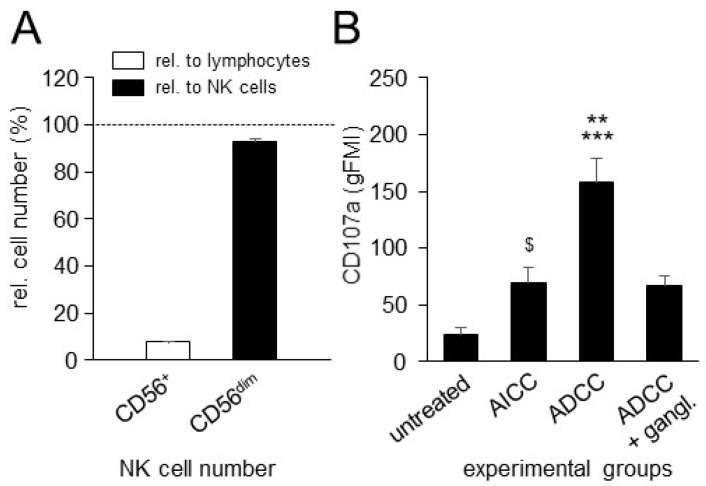
Analysis of NK-cell numbers in the samples used and impact of ADCC mediated by DB on NK-cell activity. (**A**) Analysis of NK-cell numbers and numbers of CD56dim NK cells in the effector cell samples used was performed by flow cytometry. Data are presented in % as numbers of all NK cells (white column) and CD56dim NK cells (black column), relative to all lymphocytes and all NK cells found in the samples used, respectively. (**B**) Impact of ADCC mediated by DB on expression of CD107a. ADCC was induced by incubation of the GD2-expressing NB cells LAN-1 with leukocytes of healthy donors and DB. Untreated NB cells (untreated) and NB cells incubated with effector cells without DB (AICC) served as controls. Data are shown as mean values ± SEM of at least 3 independent experiments. *t*-test; $ *p* < 0.05 vs. untreated, *** *p* < 0.001 vs. untreated, ** *p* < 0.01 vs. AICC.

**Figure 4 cancers-15-03317-f004:**
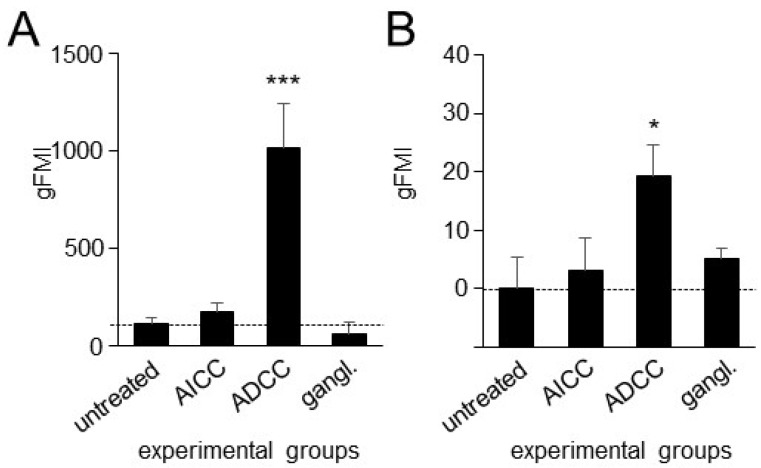
Impact of ADCC on expression of PD-L1 and PD-1 on tumor and NK cells, respectively. Effects of ADCC mediated by DB on expression levels of PD-L1 on tumor cells (**A**) and PD-1 on NK cells (**B**) were assessed using flow cytometry. ADCC was induced by incubation of the GD2-expressing NB cells LAN-1 with effector cells and DB. Untreated NB (untreated), or effector cells and NB cells incubated with effector cells without DB (AICC), served as controls. To confirm the GD2 specificity of ADCC, tumor and effector cells were incubated with DB in combination with the anti-idiotype Ab ganglidiomab (gangl.). Data are shown as mean values ± SEM of at least 3 independent experiments (**A**). *t*-test; *** *p* < 0.001 vs. AICC and untreated. (**B**). *t*-test; * *p* < 0.05 vs. AICC and untreated.

**Figure 5 cancers-15-03317-f005:**
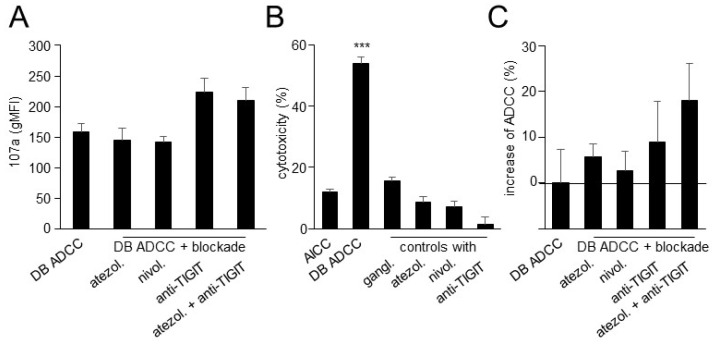
Effects of TIGIT blockade in combination with blockade of the PD-1/PD-L1 pathway on NK-cell activity and ADCC mediated by DB. (**A**) Effects of DB-ADCC in combination with immune checkpoint blockade on NK-cell activity were assessed by flow cytometry analysis of CD107a. ADCC was induced by incubation of the GD2-expressing NB cells LAN-1 with effector cells and DB. Blockade of PD-L1, PD-1 and TIGIT was performed using the anti-PD-L1 Ab nivolumab (nivol.), anti-PD-1 Ab atezolizumab (atezol.) and anti-TIGIT Ab (anti-TIGIT), respectively. (**B**) Effects of therapeutic Ab on cytotoxic activity of effector cells against NB cells. NB cells incubated with effector cells without Ab (AICC) served as controls. NB cells and effector cells incubated with DB (DB-ADCC), atezolizumab (atezol.), nivolumab (nivol.) or anti-TIGIT Ab (anti-TIGIT). To confirm the GD2 specificity of DB-ADCC, tumor and effector cells were incubated with DB in combination with the anti-idiotype Ab ganglidiomab (gangl.). (**C**) Effects of immune checkpoint blockade on ADCC mediated by DB. Data are presented in % as difference between the experimental cytotoxicity and DB-ADCC. Solid line indicates differences between experimental groups and DB-ADCC. Data are shown as mean values ± SEM of at least 3 independent experiments. (**B**) ANOVA; *** *p* < 0.001 vs. AICC.

**Figure 6 cancers-15-03317-f006:**
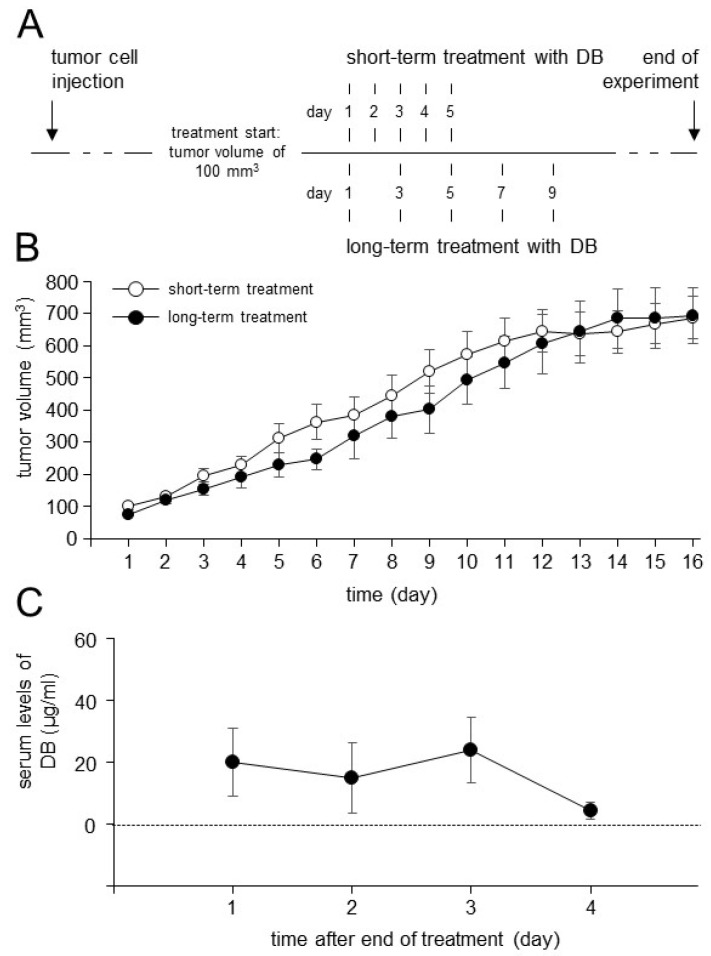
Schematic overview of the short- and long-term treatment protocols and analysis of tumor growth and DB serum concentrations. (**A**) Schematic overview of the short- and long-term treatment protocols. Mice were treated with DB either every day for five days (short-term treatment) or every second day for ten days (long-term treatment). When tumors developed a tumor size of 100 mm^3^, treatment was started. (**B**) Analysis of tumor growth in mice treated with DB given either every day for five days (short-term treatment; open circles) or every second day for ten days (long-term treatment; closed circles). Tumor growth was determined daily. When mice were sacrificed ahead of schedule due to tumor burden, the last measurement was included in the calculation of tumor growth at subsequent time points. Data are given as mean + SEM. (**C**) Serum levels of DB in mice treated with DB every second day for ten days (long-term treatment). Samples collected on days 1, 2, 3 and 4 after the end of DB treatment were assessed with ELISA as described in the Section 2. Data are given as mean + SEM.

**Figure 7 cancers-15-03317-f007:**
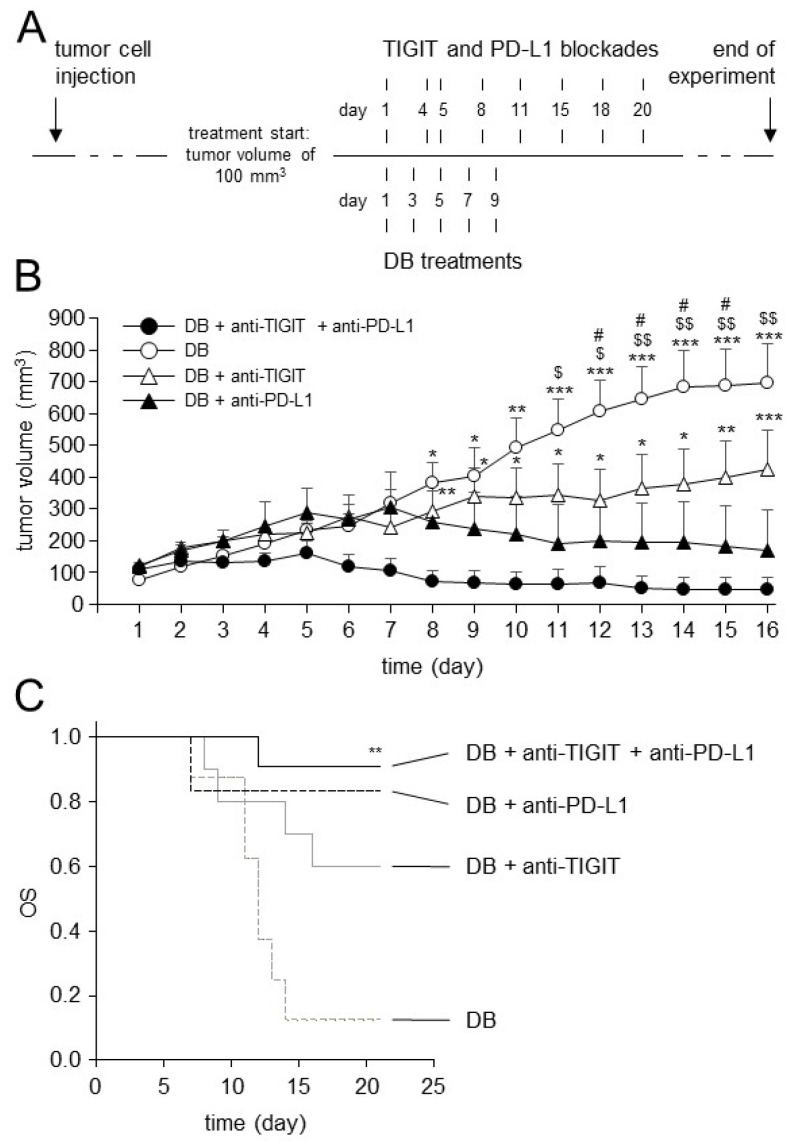
Schematic overview of the treatment protocol and analysis of tumor growth and OS in treated mice. (**A**) Schematic overview of the treatment protocol. The murine syngeneic GD2-expressing NB cells NXS2-HGW were injected subcutaneously followed by establishment of primary tumors. When tumor size of 100 mm^3^ was reached (indicated as day 1), treatments were started. (**B**) Analysis of tumor growth in treated mice. Mice received DB in combination with anti-TIGIT and anti-PD-L1 Ab (closed circles). Control mice received either a single-agent treatment with DB (open circles), a combination of DB and anti-TIGIT Ab (open triangles), or DB and anti-PD-L1 Ab (closed triangles). Tumor growth was determined daily. When mice were sacrificed ahead of schedule due to tumor burden, the last measurement was included in the calculation of tumor growth at subsequent time points. Data are given as mean + SEM. ANOVA; *** *p* < 0.001 vs. DB + anti-TIGIT + anti-PD-L1, ** *p* < 0.01 vs. DB + anti-TIGIT + anti-PD-L1, * *p* < 0.05 vs. DB + anti-TIGIT + anti-PD-L1, $$ *p* < 0.01 vs. DB + anti-PD-L1, $ *p* < 0.05 vs. DB + anti-PD-L1, # *p* < 0.05 vs. DB + anti-TIGIT; *t*-test; *** *p* < 0.001 vs. DB + anti-TIGIT + anti-PD-L1, ** *p* < 0.01 vs. DB + anti-TIGIT + anti-PD-L1, * *p* < 0.05 vs. DB + anti-TIGIT + anti-PD-L1. (**C**) Analysis of OS probabilities in mice treated with DB in combination with blockade of TIGIT and PD-L1. Mice were treated with DB in combination with anti-TIGIT and anti-PD-L1 Ab (black solid line). Control mice received either a single-agent treatment with DB (grey dashed line), a combination of DB and anti-TIGIT Ab (grey solid line), or DB and anti-PD-L1 Ab (black dashed line). Statistical analysis was performed using LogRank test, multiple comparison was carried out using Holm–Sidak method. ** *p* < 0.01 vs. DB.

**Figure 8 cancers-15-03317-f008:**
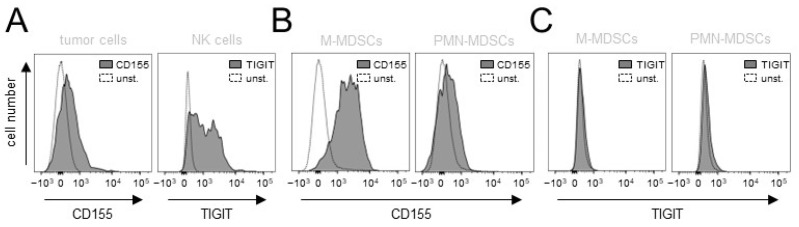
Analysis of expression of CD155 and TIGIT in primary tumor tissue. (**A**) Representative histograms of the flow cytometry analysis of CD155 and TIGIT abundance on tumor (GD2+/CD45−) and NK cells (GD2−/CD45+/CD56+) in the murine primary tumor tissue. Cells were stained with anti-murine anti-CD155 Ab and anti-murine anti-TIGIT Ab for tumor and NK cells, respectively (black-filled curve). Unstained cells served as negative controls (unst., grey curve). (**B**) Representative histograms of the flow cytometry analysis of CD155 abundance on two populations of tumor-infiltrating MDCSs (M-MDSCs, GD2−/CD45+/CD11b+/Ly6Chigh/Ly6G− and PMN-MDSCs GD2−/CD45+/CD11b+/Ly6Clow/Ly6G+). Cells were stained with anti-murine anti-CD155 Ab (black-filled curve). Unstained cells served as negative controls (unst., grey curve). (**C**) Representative histograms of the flow cytometry analysis of TIGIT abundance on two populations of tumor-infiltrating M- and PMN-MDSCs. Cells were stained with anti-murine anti-TIGIT Ab (black-filled curve). Unstained cells served as negative controls (unst., grey curve).

## Data Availability

The data of this study are available from the corresponding author upon reasonable request.

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
