# Peer review of "Combined Blockade of TIGIT and PD-L1 Enhances Anti-Neuroblastoma Efficacy of GD2-Directed Immunotherapy with Dinutuximab Beta"

_cancers, 2023, doi:10.3390/cancers15133317_

Round 1

Reviewer 1 Report

Siebert N et al. present an elegant study exploring mechanisms of anti-GD2 Ab resistance in the treatment of neuroblastoma. The way NK cells contribute to ADCC with anti-GD2 Ab has not been extensively explored although it is recognized to be important. Authors present convincing data that regarding immune checkpoint TIGIT as one of the mechanisms involved in NK cell mediated ADCC. Immunotherapy with the anti-GD2 Ab dinutuximab beta induce overexpression of PD1 and TIGIT immune checkpoints in effector cells including NK cells. Blocking of both systems results in vitro but more important in vivo in increased ADCC and anti-tumor activity when dinutuximab beta is tested.  Overall, this is a study that provides novel information with potential translational therapeutic value. 

Minor comments,

1. come typos/gramatical errors throughout the manuscript: line 77; 96; 426; 543.

2. In the M&M describe the NB resistant model to DB (not only reference)

3. In the discussion: please comment further on why in vitro results are weaker (not significant) than in vivo results for the double and single blockades.

Thanks for this revealing study.

Reviewer 2 Report

Improving anti-GD2 immunotherapy by dual blockade of TIGIT/PD-L1 is of clinical interest, and potentially addresses an area of unmet clinical need. The results presented are interesting, and of potentially of clinical relevance, but the quality of the data presented requires improvement. Many of the in vitro results  do not show significance, and the in vivo experiments requires more robust controls to allow interpretation.  Specific points:

i) Fig. 1; no control antibodies are used for comparison, 'unstained' samples are not adequate comparison as neuroblastoma cells may be 'sticky' and exhibit background non-specific binding

ii) Fig. 2; again, an isotype control antibody would be more appropriate than 'untreated. The experiment should also be repeated on more than 1 cell line, particularly given the wide SEM. It is not clear if different donors are used for each experiment.  It would be helpful to see some raw data, rather than just increase in expression. Is the term 'relative gFMI'  appropriate, when what is reported is experimental gFMI - gFMI of control, relative implies a fold change?

iii) Fig. 4; First sentence of legend say 'effector and NK cells respectively', should this be tumour and NK cells respectively? Last sentences of legend contain details about FAP - is this an error, it seems out of context? 

iv) Pg 10, line 396/397; Atezolizumab targets PD-L1, and Nivolumab targets PD-1 (opposite to written)

v)Pg 10; line 401; the authors should be cautious about stating that blockade of TIGIT increased NK activity, given the non-significant change, and that expression of CD107a is not a truly functional assay.

vi) Fig 5C would be clearer shown as just the % cytotoxicity for each treatment, rather than increase in toxicity

vii) it would be good to refer to all antibodies consistently by target (e.g anti-TIGIT) or name (e.g Nivolumab), rather than use names for some and targets for others. 

viii) Fig 6B; There is no control antibody/untreated mice; so it is not possible to say there is any anti-tumour efficacy.

ix) Fig 7; groups of mice treated with anti-TIGIT+/- anti-PD-L1 alone (without DB) should be included; it is not clear if the therapeutic effects seen are just the results of checkpoint blockade and what anti-GD2 is contributing.  Although single/dual agent checkpoint blockade therapy does not show efficacy in patients with neuroblastoma, it does show efficacy in some pre-clinical neuroblastoma models, depending on the immunogenicity of the model. If the checkpoint blockade alone does provide effective therapy, I would question whether this is a representative model.

Minor typos only require correction

Round 2

Reviewer 2 Report

Overall this is improved in response to the previous comments.